# Polyurethane Acrylate Oligomer (PUA) Microspheres Prepared Using the Pickering Method for Reinforcing the Mechanical and Thermal Properties of 3D Printing Resin

**DOI:** 10.3390/polym15214320

**Published:** 2023-11-03

**Authors:** Xiaoliang Zhao, Hua Jiao, Bin Du, Kang Zhao

**Affiliations:** 1School of Materials Science and Engineering, Xi’an University of Technology, Xi’an 710048, China; huaj@xaut.edu.cn; 2School of Materials Science and Engineering, Xi’an Polytechnic University, Xi’an 710048, China; dubin@xpu.edu.cn; 3Shaanxi Province Key Laboratory of Corrosion and Protection, Xi’an University of Technology, Xi’an 710048, China

**Keywords:** Pickering, PUA microspheres, 3D printing, thermal stability, mechanical property

## Abstract

Some photosensitive resins have poor mechanical properties after 3D printing. To overcome these limitations, a polyurethane acrylate oligomer (PUA) microsphere was prepared using the Pickering emulsion template method and ultraviolet (UV) curing technology in this paper. The prepared PUA microspheres were added to PUA-1,6-hexanediol diacrylate (HDDA) photosensitive resin system for digital light processing (DLP) 3D printing technology. The preparation process of PUA microspheres was discussed based on micromorphology, and it was found that the oil-water ratio of the Pickering emulsion and the emulsification speed had a certain effect on the microsphere size. As the oil-water ratio and the emulsification speed increased, the microsphere particle size decreased to a certain extent. Adding a suitable proportion of PUA microspheres to the photosensitive resin can improve the mechanical properties and thermal stability. When the modified photosensitive resin microsphere content was 0.5%, the tensile strength, elongation at break, bending strength, and initial thermal decomposition temperature were increased by 79.14%, 47.26%, 26.69%, and 10.65%, respectively, compared with the unmodified photosensitive resin. This study provides a new way to improve the mechanical properties of photosensitive resin 3D printing. The resin materials studied in this work have potential application value in the fields of ceramic 3D printing and dental temporary replacement materials.

## 1. Introduction

Three-dimensional printing technology is an advanced manufacturing technology that has developed rapidly in recent years and has been widely applied in various industries, such as aerospace [1], automotive [2,3], energy [4,5], medical devices [6,7], and construction [8]. Compared with traditional material-reduced manufacturing technology, 3D printing technology has the advantages of high processing flexibility, moldable products with complex structures, and low energy consumption [9,10,11]. Stereolithography has been widely applied in the industrial field, and is considered to be the earliest and most mature 3D printing technology [12]. Digital light processing (DLP) is widely used; it uses projectors as ultraviolet radiation, and photocuring resin to realize 3D printing; it is an additive manufacturing method with fast printing speed, high resolution, and the ability to prepare complex parts [13,14].

Photocuring 3D printing still has some problems, and its wide application is limited due to the fast curing speed and high cross-linking density of photocurable resin materials, resulting in insufficient mechanical properties of 3D printing photosensitive resin. Therefore, it is necessary to improve the mechanical properties of photosensitive resins. The mechanical properties of photosensitive resin can be improved by adding some reinforcing phases to the material and reacting with the resin through some functional structures. At present, the mechanical properties of photosensitive resins can be improved by three methods: (1) changing the chemical structure of the resin and increasing the rigidity of the chain, such as introducing aromatic heterocycles [15]; (2) copolymerization reinforcement, i.e., copolymerization with hard monomers [16]; and (3) blending reinforcement, i.e., blending with reinforcing materials such as SiO_2_, Al_2_O_3_, carbon nanotubes, microspheres, and GO [17,18,19,20,21].

The introduction of microspheres into 3D printing research is mainly based on their internal structure and thermal expansion properties. Wang et al. [22] used thermally expandable microspheres as an additive to the matrix, and fabricated parts using fused deposition modelling (FDM). The printed parts were subjected to heat treatment to improve mechanical properties and eliminate the clear gap between the deposition lines, which improved tensile strength and compressive strength. However, because thermoplastic polyethylene wax is used in FDM, the compressive strength is only 5.4 MPa. Wang et al. [20] fabricated glass microsphere-reinforced composites via DLP, and investigated the effects of surface chemistry and interfacial interactions on the mechanical and thermal properties of additively manufactured composites. However, composite materials that covalently combine glass functionalization with resin showed only a moderate improvement in mechanical properties, while composite materials of MAPTES (methacryloxypropyltriethoxysilane)-functionalized glass microspheres had the highest fracture strength, but only 2.28 MPa. Although these methods are promising in various fields, their applications are limited due to low mechanical strength [20,22,23]. In addition, there have been few reports on the preparation of microspheres using the Pickering emulsion template method and UV curing technology to enhance photosensitive resins.

In this paper, PUA microspheres were prepared using the Pickering emulsion method and UV curing technology. Meanwhile, PUA microspheres were added to 3D printing photosensitive resin for forming modification research. The influencing factors of the PUA microsphere preparation process were investigated, and their effects on the mechanical properties and thermal stability of the photosensitive resin 3D printing system were explored in depth. This study provides a new way to improve the mechanical properties of photosensitive resin 3D printing. The research results of this work have potential application value in the fields of ceramic 3D printing and dental temporary replacement materials.

## 2. Materials and Methods

### 2.1. Materials

Glycidyl methacrylate (GMA), Tetraethyl orthosilicate (TEOS), Gamma Valerolactone (GVL), and γ-Methacryloxypropyltrimethoxysilane (KH-570) were obtained from Macklin Chemical Reagent, Shanghai, China. Polyurethane acrylate oligomer (PUA, YC3106), 1,6-hexanediol diacrylate (HDDA), and Ethyl phenyl(2,4,6-trimethylbenzoyl)phosphinate (TPO-L) were obtained from Shanghai Yinchang New Materials, Shanghai, China. 2-hydroxy-2-methylpropiophenone (1173) was obtained from BASF, Heidelberg, Germany.

### 2.2. Preparation of PUA Microspheres

SiO_2_: SiO_2_ was prepared using the sol–gel method. First, an ammonia–water–ethanol solution was prepared, using 0.5 mol/L ammonia water, with a water concentration 5 mol/L, and making up the volume with anhydrous ethanol 200 mL. Magnetic stirring was performed at 35 °C for 1 h, then 8 mL of tetraethyl orthosilicate was dropped, a reaction was carried out at 35 °C for 24 h, the mixture was cleaned with anhydrous ethanol, centrifuged 3 times, rotated at 10,000 r/min for 10 min, and dried in a vacuum oven at 50 °C for 24 h to obtain SiO_2_.

SiO_2_ functionalization: A large number of hydroxyl groups on the surface of SiO_2_ were used to react with KH-570 for functional modification. Firstly, KH-570 was hydrolyzed, and a certain amount of solution was prepared with anhydrous ethanol and deionized water in a volume ratio of 3:2. Acetic acid was slowly added to adjust the pH value of the solution to 4. Then, KH-570 with a mass fraction of 6% SiO_2_ was added to a 100 mL mixed solution, and stirred at room temperature for 5 h to hydrolyze KH-570. After hydrolysis, 10 g of SiO_2_ was added to the above-mentioned KH-570 hydrolysis mixture solution, which was then heated to 70 °C, and allowed to react for 30 h. The prepared ethanol–deionized water mixture was washed and centrifuged 3 times at 10,000 rpm for 10 min to remove unreacted KH-570 and its by-products. The product was vacuum-dried at 70 °C for 5 h, and ground to obtain white powder SiO_2_-CH=CH_2_.

PUA microspheres: Referring to the research of Wang [24], and formulating the process parameters as shown in Table 1, a certain amount of GVL, GMA, PUA, 1173, and SiO_2_-CH=CH_2_ were mixed uniformly by magnetic stirring to prepare the Pickering emulsion oil phase. PVA aqueous solution was used as the water phase, and the oil phase and water phase were mixed in a certain ratio (oil:water, 1 mL:20 mL, 2 mL:20 mL, 3 mL:20 mL, 4 mL:20 mL, and 5 mL:20 mL) and emulsified using a high-speed disperser (T18 Digital, IKA, Staufen, Germany) at different emulsifying speeds (10 K rpm, 15 K rpm, 20 K rpm, and 25 K rpm) to prepare a series of O/W Pickering emulsions. Then, the emulsion was poured into a Petri dish and quickly cured by UV light in a closed UV light box (wavelength 365 nm, vertical distance 20 cm); the UV curing time was 5 min, 10 min, or 15 min. After light exposure, the emulsion was centrifuged, and dried to obtain a series of PUA microspheres. Meanwhile, according to Table 2, the process of PUA microspheres prepared using the Pickering emulsion template method and UV curing technology was studied, and the effects of the oil/water phase ratio, emulsification speed, emulsification time, and UV curing time on the PUA microspheres were discussed.

### 2.3. Forming of 3D Printing Photosensitive Resin Modified with PUA Microspheres

In Figure 1, the PUA microspheres are added in a certain proportion (0%, 0.1%, 0.2%, 0.3%, 0.4%, and 0.5%, mass ratio) to the photosensitive resin system for modification studies. The photosensitive resin was composed of PUA oligomer, HDDA monomer, and TPO-L photoinitiator (50:50:3, mass ratio). The modified photosensitive resin was formed into parts using a DLP 3D printer (J2-D100T-CERAMICS type, Guangdong Junjing Technology, Foshan, China), followed by post-curing (j-UV-365/405 UV curing box, Guangdong Junjing Technology, Foshan, China), and, finally, a series of PUA microsphere modified photosensitive resin 3D printing products were obtained. The light source for this experiment was 405 nm, and the printing layer thickness was 80 μm, with a single-layer exposure time of 1.5 s, a post-curing process of 405 nm, a power of 4 MW/cm^2^, and an irradiation time of 2 min.

### 2.4. Characterization Techniques

Fourier-transform infrared (FTIR) spectroscopy was measured using an FTIR spectrometer (Nicolet IS50 type by Thermo, Waltham, MA, USA) in the wave number range from 4000 to 400 cm^−1^. Thermogravimetric analysis (TGA) of SiO_2_, SiO_2_-CH=CH_2_, PUA microspheres, and PUA microsphere-modified photosensitive resin was performed using a Q 500 apparatus (TA Instruments, New Castle, DE, USA). A heating rate of 15 °C/min under a nitrogen atmosphere (30 mL/min) was chosen, and the temperature ranged from 30 to 600 or 1000 °C. Differential scanning calorimetry (DSC) measurements were performed on a Q 2000 apparatus (TA Instruments, New Castle, DE, USA). A heating rate of 15 °C/min was used under a nitrogen atmosphere (30 mL/min). The temperature range was −50 to 350 °C. The SiO_2_, SiO_2_-CH=CH_2_, morphology of PUA microspheres, and cross-sectional microstructure of PUA microsphere-modified photosensitive resin were observed using a scanning electron microscope (SEM, Quanta-450-FEG, FEI, Hillsboro, OR, USA). The particle size and distribution of SiO_2_ and SiO_2_-CH=CH_2_ were investigated using the Malvern nanoparticle size analyser (Malvern Nano ZS90, Worcestershire, UK). The mechanical properties were measured using a universal testing machine (UTM5504, Shenzhen Sansi Testing Equipment, Shenzhen, China), following the tensile ISO-14704:2016 [25] and bending GB/T 9341-2008 [26] standards, with a tensile rate of 5 mm/min and a bending speed of 1 mm/min; each group was tested 5 times.

## 3. Results and Discussion

### 3.1. PUA Microspheres

The preparation process and principle of PUA microspheres are shown in Figure 2a. Firstly, step 1 is used to prepare SiO_2_ [27], using the sol–gel method. Tetraethyl silicate is dissolved in anhydrous ethanol solvent, hydrolyzed into sol, then slightly cross-linked into a loose gel network, formed, dried to remove water and ethanol, and further converted into a deeply cross-linked structure to obtain SiO_2_. The microstructure of SiO_2_ in Figure 2b is relatively uniform in size, and the measurement is about 200 nm. Meanwhile, the particle size and size distribution in Figure 2f are the same as those observed using SEM, the particle size distribution shows a single dispersion, and the Z-average diameter of silica particles is 196 nm. In Figure 2g, the infrared spectrum of SiO_2_ shows that 3400 cm^−1^ is the absorption peak of the OH antisymmetric stretching vibration on the surface of silica; the strong and broad absorption band of 1104 cm^−1^ is the Si-O-Si antisymmetric stretching vibration peak of SiO_2_ [28]; 800 cm^−1^ is the symmetric stretching vibration peak of the Si-O bond, and 471 cm^−1^ is the flexural vibration peak of the Si-O bond. Therefore, spherical silica has been successfully prepared in step 1.

In Figure 2a, step 2 is the hydrolysis reaction of the SiO_2_ double-bond functional modifier silane coupling agent KH-570. When 1 mol of KH-570 reacts with 3 mol of H_2_O, the siloxane group [-Si-OR] in the molecular structure of KH-570 combines with the hydrogen atom in the water molecule to form the silanol group [-Si-OH], which is unstable, and then [29], it reacts with -OH on the surface of silica to form a SiO_2_-O-Si-C bond (step 3). KH-570 is anchored to the surface of SiO_2_ particles through covalent bonds (SiO_2_-O-Si), and the SiO_2_-functionalized product is labelled as SiO_2_-CH=CH_2_. The SEM of SiO_2_-CH=CH_2_ is shown in Figure 2c, which shows that the particle size is uniform and non-adherent, and the measurement is about 200 nm. The particle size and size distribution of (f) in Figure 2 are the same as those observed using SEM, showing a uniform size and single dispersion. The Z-average diameter of the silica particles is 188 nm. Compared to unmodified SiO_2_, the particle size decreases and the particle size distribution narrows. This is because the functionalized silica surface is grafted with some KH-570, resulting in a decrease in the degree of aggregation between microspheres and an increase in hydrophobic dispersion [30]. In Figure 2g, the SiO_2_-CH=CH_2_ infrared spectrum shows the typical group absorption peak of SiO_2_, as described above. Meanwhile, the -CH_3_ tensile vibration absorption peak and the C-H stretching vibration peak are found at 2927 cm^−1^ and 2855 cm^−1^, respectively. The Si-O-Si antisymmetric tensile vibration peak at 1104 cm^−1^, the Si-O bond symmetric stretching vibration peak at 800 cm^−1^, and the Si-O bond bending vibration peak at 471 cm^−1^ become weaker, which proves that the silica modification is successful, and KH-570 is successfully anchored on the surface of SiO_2_ particles.

In step 4, PUA microspheres were prepared using Pickering emulsion as a template and UV curing technique. First, the Pickering emulsion was prepared by adding PVA as a co-stabilizer to the water phase, PUA and GMA as interfacial crosslinking reactants, GVL as co-solvent, and 1173 as photoinitiator. SiO_2_-CH=CH_2_ particles are particle emulsifiers that make the Pickering emulsion stable. In addition, SiO_2_-containing double bonds on the surface can improve its compatibility in the oil phase PUA, and can also participate in the photocuring reaction at the interface. The Pickering emulsion was prepared by a high-speed dispersion of the water and oil phases. Finally, a UV photopolymerization reaction occurred at the oil–water interface to form PUA microspheres after rapid illumination using a 365 nm UV LED lamp. However, the mechanically stirred lotion presents water-in-oil-in-water double emulsions. The water phase inside the droplets migrates outwards in the oil phase and coalesces. If the photopolymerization rate is slower than the phase separation rate, a spherical water-rich phase is formed in the oil phase, and a hollow microsphere structure is prepared. If the photopolymerization rate is faster than the rate of phase separation rate, a porous or solid structure is formed [31]. In this experiment, the Pickering emulsion was prepared using rapid UV photopolymerization, so the preparation was mostly porous or solid PUA microspheres. Figure 2d shows the SEM of a single PUA microsphere, where the surface of the microsphere shell is surrounded by silica particles, and the shell layer polymerized by PUA and GMA also contains silica particles. From the SEM image of the microsphere fragmentation in Figure 2e, it can be seen that the PUA microsphere is composed of some small PUA microspheres, silica particles, and the polymer reacted by PUA and GMA, with small pores in the middle. This is because the Pickering emulsion is prepared for rapid UV photopolymerization, and the photopolymerization rate is faster than the phase separation rate, so most of the prepared PUA microspheres are porous. In Figure 2g, the infrared spectra of PUA microspheres showed some characteristic peaks of SiO_2_-typical groups, while the typical characteristic absorption peaks of polyurethane acrylate also appeared. In particular, 3425 cm^−1^ is the N-H bond stretching vibration peak in urethane, 1731 cm^−1^ is the carboxyl-COO- group vibration peak in polyurethane acrylate, 1181 cm^−1^ is the C-O-C group stretching vibration peak in urethane, and 1610 cm^−1^ was the unreacted C=C group stretching vibration peak in PUA and GMA, indicating that PUA microspheres were successfully prepared.

The process of PUA microspheres prepared using the Pickering emulsion template method and UV curing technology was studied, and the effects of the oil/water phase ratio, emulsification speed, emulsification time, and UV curing time on the microstructure, and the particle size distribution of PUA microspheres were discussed. The SEM is shown in Figure 3, and the particle size distribution is shown in Figure 4.

Figure 3 and Figure 4 show that the size distribution of the PUA microspheres prepared using the Pickering lotion template method and UV curing technology are different. In Figure 3, A-1, A-2, A-3, A-4, and A-5 are PUA microspheres prepared under different oil/water phase ratios. From the microscopic morphology, it can be seen that as the oil/water phase ratio increases, the particle size of the PUA microspheres decreases. B-1, B-2, and B-3 are PUA microspheres prepared under different dispersion times, and it can be seen that the dispersion time has little effect on the appearance and particle size of the microspheres. C-1, C-2, and C-3 are PUA microspheres prepared under different lighting times. It can be seen that the lighting time has little effect on the appearance and particle size of the microspheres. D-1, D-2, D-3, and D-4 are PUA microspheres prepared at different rotation speeds. It can be seen that the particle size of the microspheres gradually decreases with the increase in dispersion speed, which is because the increase in emulsifying shear rotational speed reduces the probability of droplet aggregation of the Pickering emulsion. Furthermore, a particle size consistent with SEM is obtained from the particle size test in Figure 4. Therefore, in the process study of PUA microspheres, the oil/water phase ratio of the Pickering emulsion and the emulsification speed have a certain influence on the size of the PUA microspheres.

In Figure 3, E is a PUA microsphere particle; the microsphere is relatively regular, and its shell is covered by relatively uniform particles of dioxide. The reason for the formation of such a structure is that when silica is pre-dispersed in the oil phase, its surface is first completely wetted by the oil phase mixture, and its volume in the oil phase is much larger than that in the water phase at the water–oil interface. The photosensitive reactants in the oil phase near the water–oil interface occur rapidly during the photopolymerization process, and can be completed in just over ten seconds or even a few seconds [24]. A portion of the volume of the SiO_2_-CH=CH_2_ particles is squeezed from the oil phase to the water phase. Therefore, the special morphology of SiO_2_-CH=CH_2_ particles is formed on the surface of the hollow microspheres.

### 3.2. Thermal Properties of SiO_2_, SiO_2_-CH=CH_2_, and PUA Microspheres

As the TGA of SiO_2_ and SiO_2_-CH=CH_2_ in Figure 5a shows, there is thermal decomposition in the temperature range of 30–200 °C, which is caused by the evaporation of water molecules adsorbed on the surface of silica. The thermal decomposition in the temperature range of 200–600 °C in the thermogravimetric curve of SiO_2_-CH=CH_2_ is caused by the decomposition of KH-570 grafted onto the SiO_2_ surface modification. The weight loss rate of SiO_2_-CH=CH_2_ at 1000 °C is 15.65%, and the weight loss rate of unmodified SiO_2_ at 1000 °C is 12.55%, an increase of 3.1%. This indicates a successful functionalization modification of SiO_2_ by KH-570.

As the TGA of the PUA microspheres in Figure 5a shows, at 30–300 °C, the thermal decomposition of the PUA microspheres is slow, mainly caused by the pyrolysis of some small molecules. At 300–500 °C, the microspheres undergo significant thermal decomposition, mainly caused by the breakage of the polymer molecular chain formed by the cross-linking reaction between PUA and GMA. At 500–1000 °C, the pyrolysis quality of the microspheres remains unchanged; this portion consists mainly of carbon and SiO_2_ obtained from material pyrolysis. In Figure 5a, it can be observed that the weight of the SiO_2_ sample gradually decreases with increasing temperature, leading to a weight loss of 14% at 1000 °C. The weight loss mainly includes two parts: one is the removal of water adsorbed on SiO_2_ before 150 °C, and the other is the further dehydration and condensation reaction between adjacent silicon hydroxyl groups on the surface of SiO_2_ after 150 °C. As the DSC curve of the PUA microspheres in Figure 5b shows, the glass transition temperature of polyurethane acrylate is 64.33 °C.

### 3.3. PUA Microsphere-Modified 3D Printing Photosensitive Resin

The 3D printing forming principle of the photosensitive resin modified with PUA microspheres is shown in Figure 6a. A modified photosensitive resin was prepared by mixing PUA oligomer, HDDA monomer, TPO-L initiator, and other photosensitive resin components with PUA microspheres. The SOLIDWORKS v2017 software was used to draw the model of the printed part and save it as STL, loaded into the DLP printer’s computer control system. The DLP 3D printer is a bottom-exposure digital-light-processing rapid prototyping method that uses a digital light source in the form of surface exposure to project layer by layer onto the contact surface between the liquid photosensitive resin and the storage tank, and solidify layer by layer for molding. The prepared modified photosensitive resin is poured into the storage pool, with the printing plane firmly attached to the surface of the photosensitive resin. The light source under the printer illuminates the entire cross-section of the designed model for curing. After printing one layer, the printing platform moves up a layer thickness distance along the Z-axis to continue printing. This process is repeated until printing is complete. After printing, the model is removed from the platform with a shovel, cleaned with anhydrous ethanol, and then post-cured to obtain the printed part.

Curing is a process of photopolymerization chemical reaction that involves the absorption of a photon by an initiator and the polymerization of a large number of oligomers and monomer molecules into large molecules [32]. It is a free radical photopolymerization reaction. The first step is chain initiation, and the photoinitiator breaks down into free radical molecules after irradiation, forming primary active radical initiators. The primary free radicals add to oligomers or monomers, forming oligomeric or monomeric radicals. Then, the chain grows and the oligomer or monomer free radical opens the π bond on the double bond of the oligomer or monomer, via an addition reaction, and forms a new free radical. The new free radical continues the addition reaction with the oligomer or monomer, forming more chain free radicals with structural units [33]. Finally, in chain termination, free radicals have high activity, are difficult to exist in isolation, and are easy to interact with until termination. In the free radical photopolymerization process, the initiation and termination reactions are almost simultaneous with light; that is to say, the photosensitive resin begins to polymerize when exposed to light, and when the radiation source is removed, the polymerization reaction is immediately terminated.

In Figure 6b, no particles can be seen in the photosensitive resin without PUA microspheres. The microstructure of the cross-section of the photosensitive resin with 0.5% PUA microspheres can be seen locally, as shown in Figure 6c. It can be seen from the cross-section that the distribution of the photosensitive resin is uneven. Moreover, it was found that some of the PUA microspheres embedded in the photosensitive resin were disturbed and pulled out during fracture, as shown in Figure 6g, while others were directly pulled out, as shown in Figure 6e. From Figure 6e, it can be seen that there is spherical silica with the shell layer of PUA microspheres peeling off in the pit of the detached microspheres. From the SEM of the stretched section of the photosensitive resin, it can be concluded that the PUA microspheres can consume energy during the stretching process by being dislodged or pulled out, thereby achieving the goal of improving the mechanical properties.

### 3.4. Mechanical Properties of PUA Microsphere-Modified Photosensitive Resin

From Figure 7 and Table 3, it can be concluded that as the content of PUA microspheres increases, the tensile strength and bending strength of the modified photosensitive resin gradually increase. When the modified photosensitive resin contains 0% PUA microspheres, the tensile strength is 8.13 MPa, the elongation at break is 8.93%, and the bending strength is 10.23 MPa; when the modified photosensitive resin contains 0.5% PUA microspheres, the tensile strength is 14.32 MPa, the elongation at break is 13.15%, and the bending strength is 12.96 MPa. Compared to the photosensitive resin without PUA microspheres, the tensile strength is increased by 79.14%, the elongation at break is increased by 47.26%, and the bending strength is increased by 26.69%. This indicates that the addition of PUA microspheres improves the mechanical properties of the photosensitive resin, due to the presence and surface coating of SiO_2_ inside the PUA microspheres, as SiO_2_ is an inorganic non-metallic material with excellent impact resistance and stiffness. In addition, from the SEM of the stretched sections of the photosensitive resins, shown in Figure 6e,g, it was found that some PUA microspheres were dislodged or directly pulled out, which can consume the energy applied externally during the stretching process, thereby improving the mechanical properties. Therefore, PUA microspheres can provide excellent mechanical properties for photosensitive resin materials.

### 3.5. Thermal Properties of PUA Microsphere-Modified Photosensitive Resin

From Figure 8a,b, it can be seen that as the PUA microsphere content increases, the initial decomposition temperature of the modified photosensitive resin increases, and the weight loss rate at 600 °C of the material decreases, indicating that the addition of PUA microspheres improves the thermal stability of the photosensitive resin, due to the internal presence and surface coating of SiO_2_ in PUA microspheres, as SiO_2_ is an inorganic non-metallic material. From the TG curve of SiO_2_ in Figure 5a, it can be concluded that SiO_2_ has excellent heat resistance performance. As shown in Figure 8c, when the photosensitive resin contains 0% PUA microspheres, the initial thermal decomposition temperature is 340.05 °C, and the weight loss rate is 92.43% at 600 °C. When the photosensitive resin contains 0.5% PUA microspheres, the initial thermal decomposition temperature is 376.28 °C, and the weight loss rate at 600 °C is 95.07%. Compared with the modified photosensitive resin without microspheres, the initial thermal decomposition temperature increased by 10.65%, and the weight loss rate at 600 °C decreased by 2.64%. Therefore, PUA microspheres can improve the thermal stability of photosensitive resin materials.

Figure 8d shows that the glass transition temperature and melting point of the photosensitive resin containing 0% PUA microspheres are 48.11 °C and 286.07 °C, respectively. The glass transition temperature of PUA microspheres containing 0.2% PUA is 51.23 °C, with a melting point of 311.74 °C. The glass transition temperature of PUA microspheres containing 0.5% PUA is 52.00 °C, with a melting point of 332.12 °C. It can be concluded that the addition of PUA microspheres increases the glass transition temperature and melting point of photosensitive resins, because the addition of PUA microspheres containing SiO_2_ reduces the mobility of polymer molecular chains and acts as an obstacle to chain movement, increasing the glass transition temperature. PUA microspheres increase the resistance to rotation in the polymer chain, and the conformational change during melting is small, resulting in an increase in the melting point [34,35].

## 4. Conclusions

In this paper, double-bonded SiO_2_ was prepared using the sol–gel method and KH-570 modification. This was added as an emulsifier to the oil phase of a Pickering emulsion, and PUA microspheres were successfully prepared using the Pickering emulsion template method and UV curing technology to improve the photosensitive resin for 3D printing. SEM, dynamic light scattering, and FTIR provide evidence for the formation of PUA microspheres, and the average particle size of the microspheres is 188 nm. In the preparation process of the PUA microspheres, emulsification and light time had no obvious effect on the microsphere morphology. As the oil/water ratio and emulsification speed of the Pickering emulsion increased, the particle size of PUA microspheres decreased. Benefiting from the fact that the microsphere is directly displaced or pulled out in the 3D printing photosensitive resin, the energy applied from the outside world is consumed, and the microsphere contains heat-resistant SiO_2_, the mechanical properties and thermal stability of the modified photosensitive resin are improved. Compared with a single photosensitive resin, when the amount of PUA microspheres added is 0.5%, the 3D printing photosensitive resin formed has an increase of 79.14% in tensile strength, 47.26% in elongation at break, 26.69% in bending strength, and 10.65% in initial thermal decomposition temperature. This study provides a new way to improve the mechanical properties of 3D printing photosensitive resin.

## Figures and Tables

**Figure 1 polymers-15-04320-f001:**
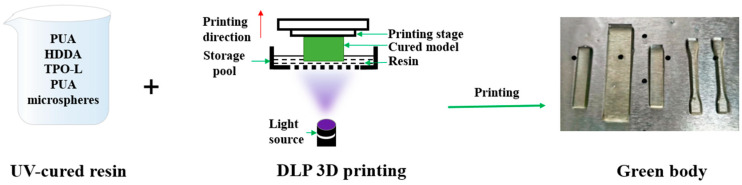
The preparation process of resin.

**Figure 2 polymers-15-04320-f002:**
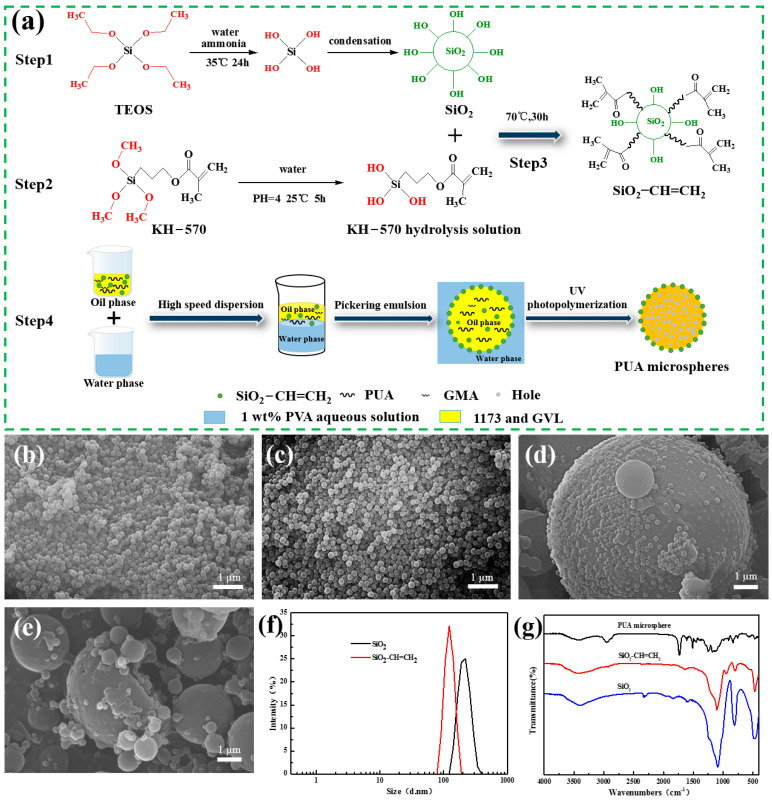
(**a**) Schematic illustration of the preparation of PUA microspheres. SEM images (**b**–**e**) of SiO_2_, SiO_2_-CH=CH_2_, PUA microspheres, and the morphology of the microspheres after grinding. (**f**) Laser diffraction image of the particle size distribution. (**g**) FTIR of SiO_2_, SiO_2_-CH=CH_2_, and PUA microspheres.

**Figure 3 polymers-15-04320-f003:**
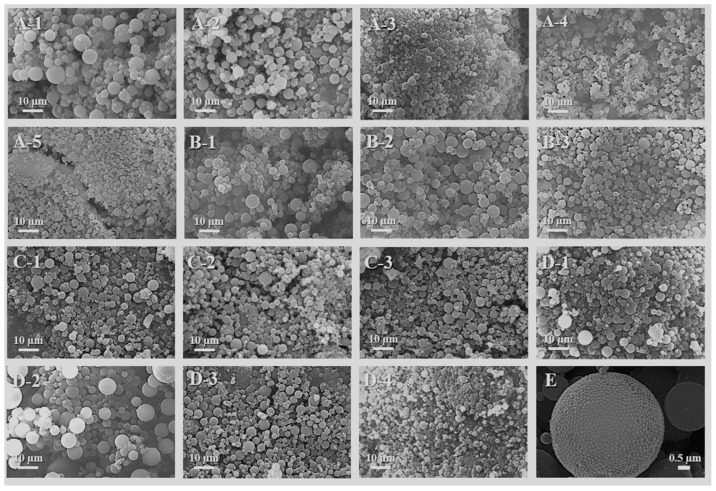
SEM of PUA microspheres under different process conditions: (**A-1**–**A-5**), oil/water phase ratio (1 mL:20 mL, 2 mL:20 mL, 3 mL:20 mL, 4 mL:20 mL, and 5 mL:20 mL). (**B-1**–**B**-**3**), emulsification time (3 min, 5 min, and 8 min). (**C-1**–**C-3**), UV curing time (5 min, 10 min, and 15 min). (**D-1**–**D-4**), emulsification speed (10 K rpm, 15 K rpm, 20 K rpm, and 25 K rpm). (**E**), a PUA microsphere.

**Figure 4 polymers-15-04320-f004:**
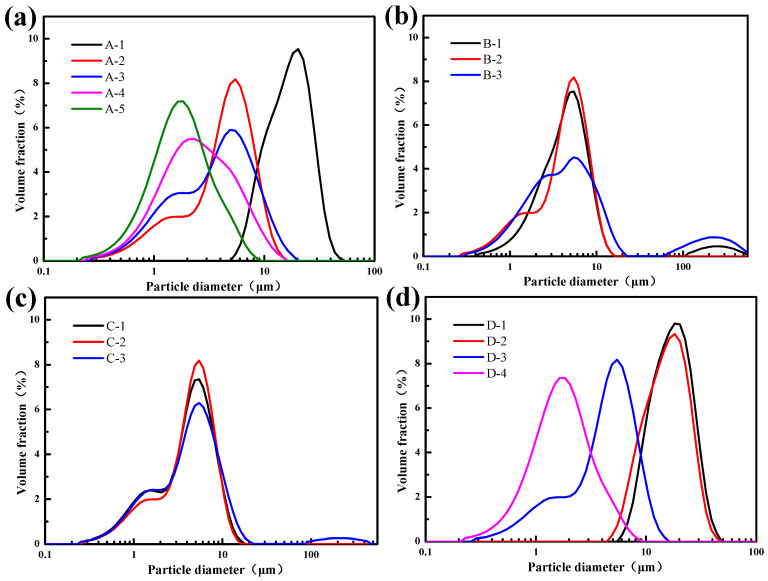
Particle size distribution of PUA microspheres under different process conditions: (**a**) oil/water phase ratio, (A-1–A-5). (**b**) Emulsification time, (B-1–B-3). (**c**) UV curing time, (C-1–C-3). (**d**) Emulsification speed, (D-1–D-4).

**Figure 5 polymers-15-04320-f005:**
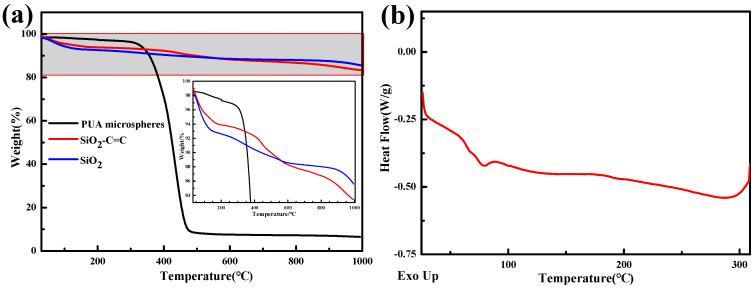
(**a**) TGA of SiO_2_, SiO_2_-CH=CH_2_, and PUA microspheres. The gray background is the range of the enlarged part in subfigure (**a**). (**b**) DSC thermograms of PUA microspheres.

**Figure 6 polymers-15-04320-f006:**
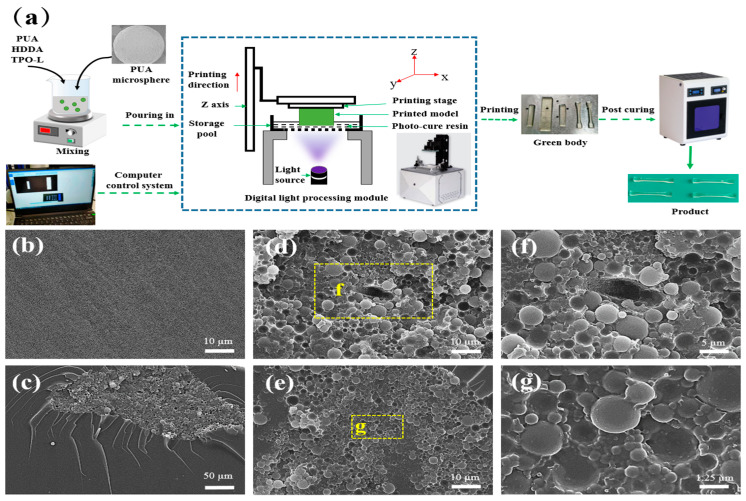
(**a**) The principle of 3D printing forming of PUA microsphere-modified photosensitive resin. (**b**–**e**) SEM of the tensile section of PUA microspheres modified photosensitive resin: (**b**) 0%, (**c**) 0.5%; (**d**,**e**) is the microscopic view of the cross-section, (**f**) is a partial enlargement of (**d**), and (**g**) is a partial enlargement of (**e**).

**Figure 7 polymers-15-04320-f007:**
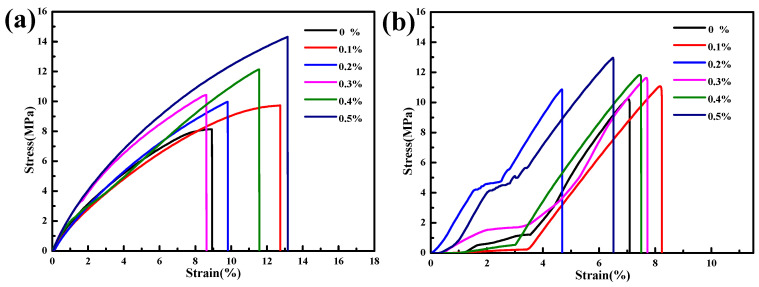
Stress-strain curves of photosensitive resin modified with different contents of PUA microspheres: (**a**) tensile, (**b**) 3-point bending.

**Figure 8 polymers-15-04320-f008:**
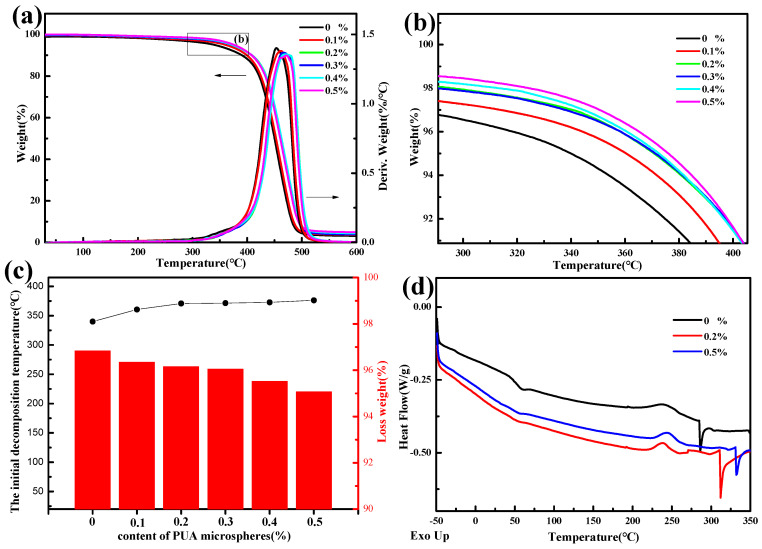
(**a**) Photosensitive resin TGA with different contents of PUA microspheres. (**b**) Drawing of partial enlargement for (**a**). (**c**) The initial decomposition temperature and weight loss rate at 600 °C of PUA microspheres with different contents of photosensitive resins. (**d**) DSC thermograms of the photosensitive resin modified with PUA microspheres.

**Table 1 polymers-15-04320-t001:** Proportions of water phase and oil phase in a Pickering emulsion.

Water Phase		Oil Phase				
Water (mL)	PVA (wt%)	SiO_2_-CH=CH_2_ (wt%)	GVL (wt%)	PUA (wt%)	GMA (wt%)	1173 (wt%)
10	1	1	25	36	36	3

**Table 2 polymers-15-04320-t002:** Process parameters for the preparation of PUA microspheres with Pickering emulsion.

Number	Oil/Water Phase Ratio (Oil:Water)	Emulsifier Concentration	Emulsification Speed	Emulsification Time	UV Curing Time	Particle Size (D_50_, μm)
A-1	1 mL:20 mL	1%	20 K rpm	5 min	10 min	18.09
A-2	2 mL:20 mL	1%	20 K rpm	5 min	10 min	4.80
A-3	3 mL:20 mL	1%	20 K rpm	5 min	10 min	4.09
A-4	4 mL:20 mL	1%	20 K rpm	5 min	10 min	2.59
A-5	5 mL:20 mL	1%	20 K rpm	5 min	10 min	1.83
B-1	2 mL:20 mL	1%	20 K rpm	3 min	10 min	4.85
B-2	same as A-2					
B-3	2 mL:20 mL	1%	20 K rpm	8 min	10 min	4.90
C-1	2 mL:20 mL	1%	20 K rpm	5 min	5 min	4.55
C-2	same as A-2					
C-3	2 mL:20 mL	1%	20 K rpm	5 min	15 min	4.78
D-1	2 mL:20 mL	1%	10 K rpm	5 min	10 min	18.11
D-2	2 mL:20 mL	1%	15 K rpm	5 min	10 min	16.52
D-3	same as A-2					
D-4	2 mL:20 mL	1%	25 K rpm	5 min	10 min	1.80

**Table 3 polymers-15-04320-t003:** Mechanical properties of photosensitive resins modified with different contents of PUA microspheres.

Sample	Tensile Strength (MPa)	Elongation at Break (%)	Bending Strength (MPa)
0%	8.13 ± 1.11	8.93 ± 0.93	10.23 ± 0.40
0.1%	9.73 ± 0.24	12.83 ± 1.24	11.08 ± 1.12
0.2%	9.97 ± 0.32	9.81 ± 0.72	10.86 ± 0.33
0.3%	10.43 ± 0.41	8.65 ± 0.53	11.63 ± 0.20
0.4%	12.14 ± 0.33	11.55 ± 0.81	11.83 ± 0.31
0.5%	14.32 ± 0.65	13.15 ± 0.72	12.96 ± 0.64

## Data Availability

Not applicable.

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
