# Peer review of "Polyurethane Acrylate Oligomer (PUA) Microspheres Prepared Using the Pickering Method for Reinforcing the Mechanical and Thermal Properties of 3D Printing Resin"

_polymers, 2023, doi:10.3390/polym15214320_

Round 1

Reviewer 1 Report

Comments and Suggestions for Authors

The manuscript described using the pickering emulsion method to prepare polyurethane microparticles to enhance the physical properties of 3D-printed polyurethane acrylate-based materials. The content in the current manuscript is too superficial and lacks some critical experiments. The data interpretation in the manuscript is also not scientifically sound. Therefore, in my opinion, I am afraid it cannot be considered for publication.

Comments:

1.      The title should be presented in a more concise form.

2.      The language of the whole manuscript must be improved.

3.      The authors used pickering emulsion method to fabricate PUA microsphere. During the procedure, SiO2-CH=CH2 were used as the solid surfactant to stabilize the PUA microsphere. This means that SiO2 is also used as a solid filler and crosslinker in the final material. It is well-known that the functional SiO2 nanopowder can be used as a filler to effectively increase the mechanical properties of materials. The authors should do a control experiment that used only SiO2-CH=CH2 as a solid filler to see how much it can enhance the mechanical properties of the original resin, and compare the results with the materials that contain PUA microspheres.

4.      The authors used several different conditions to prepare the PUA microspheres, as listed in Table 2. Then, on page 7, they roughly discussed how different parameters affect the particle size with SEM images. I would suggest adding a table or figure to show a more clear way about how each parameter affects the particle size. Or at least the particle size should be also listed in Table 2 for each set.

5.      Figure 5 only shows tensile and 3-point bending experiments for single test samples. The summary graph or table with errors or deviations should be presented.

6.      The 0.5% sample shows higher stress and strain in tensile but shows less flexibility in the 3-point bending experiment, what could be the reason?

7.      In both Figure 5a and 5b, there is no obvious trend of the mechanical properties change as the amount of PUA microspheres gradually increased, which may indicate that the data reliability is low. Again, it is important to show the statistical results rather than just the results of a single experiment.

Comments on the Quality of English Language

There are too many syntax errors.

Author Response

Dear reviewers,

Thank you so much for your email and advices about our paper on polymers entitled “PUA microspheres were prepared by the Pickering method and it reinforced the mechanical and thermal properties of photosensitive resin in 3D printing” (ID: polymers-2608505). These comments and suggestions are very valuable and helpful for revising and improving the manuscript, as well as providing good guidance for our research. We have carefully reviewed your comments and made revisions. According to which a red color revised version of the manuscript was prepared. The main modifications of this paper and the replies to the reviewer's comments are shown in the uploaded word attachment.

Thank you and best regards.

Yours sincerely

Xiaoliang Zhao

E-mail: zhaoxiaoliang@xpu.edu.cn

Reviewer 2 Report

Comments and Suggestions for Authors

Dear Authors, thank you for this interesting paper  I had an opportunity to review. I think it is signifficant in this field, but here are some suggestions of mine:

1. Line 82+, please draw a graph of how this resin was prepared so that the reader would have a "cheat sheet" of it

2. The preparation of the sample (with its UV curing) seems complicated - please, add the time needed

3. Fig. 3.3, graph a should be larger

4. The discussion should be widened. Maybe a nice aspect would be finding similar studies? Maybe other substances were added and they improved or decreased the values? You should write about perspectives. Maybe the good idea would be thinking on how the properties would change in time? What about the resins that have a cartridges (like Dental LT Clear, Formlabs) - how can you prepare those specimens? How about biocompatibility - could it or should it change? Why? Provide limitations. 

Author Response

(The authors gave the same response as above.)

Reviewer 3 Report

Comments and Suggestions for Authors

Thanks for including the detail of the tensile test detail, Figure 5 explains everthing

Comments on the Quality of English Language

some minor correction must be spotted

Author Response

(The authors gave the same response as above.)

Reviewer 4 Report

Comments and Suggestions for Authors

1. Title of the manuscript is very lengthy and vague.

2. Include the scope of the research in the abstract.

3. Introduction part need to be revised.

4. Add more details about microspheres preparation with figure for better understanding.

5. What the authors are trying to explain from Fig. 1(a)?

6. SEM images presented in Fig. 2 are not clear.

7. All line graphs are to be explained with error percentage.

Comments on the Quality of English Language

Need to be improved

Author Response

(The authors gave the same response as above.)

Reviewer 5 Report

Comments and Suggestions for Authors

Refer to the attached file.

Comments on the Quality of English Language

The English language usage need to be thoroughly checked.

Author Response

(The authors gave the same response as above.)

Reviewer 6 Report

Comments and Suggestions for Authors

The authors have presented a study that explores the development of a novel PUA microsphere to enhance the mechanical and thermal properties of 3D-printable resins. The observation that the properties of the 3D-printable resin can be enhanced with the addition of up to 0.5% PUA microsphere is noteworthy and holds significance in the realm of material science, especially in the context of 3D printing resin reinforcement. I believe that this paper has the potential to make a valuable contribution to the literature, and I recommend its acceptance after some necessary revisions. Here are my comments:

1.    Chemical Formula of KH-570:

Please provide the chemical formula for KH-570. It's mentioned for the first time on line 90, and this inclusion will offer clarity to the readers.

2.    3D Printing Parameters:

I suggest relocating the details about the 3D printing parameters found in lines 331–334 to the methods section. Specifically, "The light source for this experiment is 405 nm, with an irradiation time of approximately 2 minutes."

3.    Weight Changes of SiO2:

Regarding the TGA curve in Figure 3(a), it's observed that the weight of the SiO2 sample decreases gradually with a rise in temperature, leading to a 20% weight loss at 1000ºC. Could you elucidate the underlying cause for this behavior?

4.    Repeatability of the Mechanical Test:

Mechanical properties, like tensile strength, often exhibit variability. Consequently, it's crucial to specify the number of samples tested for each group to ascertain the mechanical properties of the materials reliably.

I offer these comments and suggestions in the hope of enhancing the robustness of the manuscript. I look forward to seeing the revised version.

Comments on the Quality of English Language

There are some typos. 

English proofreading is required.

Author Response

(The authors gave the same response as above.)

Round 2

Reviewer 1 Report

Comments and Suggestions for Authors

The authors properly addressed my comments and revised the manuscript.

Reviewer 4 Report

Comments and Suggestions for Authors

The revised manuscript is improved and the authors are responded well for the queries raised by the Reviewers. Now I recommend this manuscript for publication in this Journal.

Comments on the Quality of English Language

Minor editing of English language required

Reviewer 6 Report

Comments and Suggestions for Authors

The manuscript has been improved.